# Storing Energy from External Power Supplies Using Phase Change Materials and Various Pipe Configurations

Daniel Aprile, Samer Al-Banna, Arraventhan Maheswaran, Joshua Paquette and Mohamad Ziad Saghir *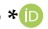

Mechanical and Industrial Engineering, Ryerson University, Toronto, ON M5B 2K3, Canada; daniel.aprile@ryerson.ca (D.A.); sam018273@gmail.com (S.A.-B.); amaheswaran@ryerson.ca (A.M.); joshua.paquette@ryerson.ca (J.P.)
* Correspondence: zsaghir@ryerson.ca

**Abstract:** Phase change materials are commonly used for energy storage. Heat transfer enhancement and heat storage are the two main goals in this paper. A cylindrical pipe covered with phase change material is investigated numerically. Ideally, a high temperature liquid flows through the pipe, resulting in heat transferred to the phase change material. To enhance the heat transfer, various configurations involving the addition of a twisted tape inside of the pipe and the use of helical shape pipes were investigated. A straight pipe with no twisted tape insert was also analyzed and used as a benchmark case. All the configurations had constant properties such as material selection, overall size, pipe diameter and inlet Reynold's number, so the performance could be compared under similar conditions. All initial configurations were simulated and the heat transfer rate, Nusselt number, friction factor and performance evaluation criterion (PEC) of the designs were determined. It was found that the heat transfer rate and Nusselt number of all the various designs yielded higher results than the reference straight pipe configuration. Additionally, due to the added complexity in the flow caused by the insert, the friction factor of all the configurations was also higher. The helical pipe configuration was the only configuration that had a PEC higher than that of the reference straight pipe. This is because the negative impacts caused by the friction factor outweighed the gains in Nusselt number for the twisted tape designs. It was also hypothesized that lowering the inner diameter of the helical pipe would increase the PEC. Further simulations with modified inner diameters were done to test the hypothesis. The simulations confirmed the hypothesis, as the pipes with inner diameters 0.75 and 0.5 cm led to a 50% and 150% increase in the PEC respectively, when compared to an inner diameter of 1 cm. It was also determined that smaller inner diameters led to lower outlet temperatures meaning a higher percentage of the thermal energy from the fluid was transferred to the phase change material.

**Keywords:** phase change material; twisted tape; helical pipe; heat transfer

## 1. Introduction

Currently one of the most popular topics in power generation is generating power using renewable sources such as solar and wind power. One of the major issues in effectively implementing these renewable energy sources is the fact that they produce power intermittently due to natural factors. To overcome this, a system can be implemented to effectively store the power generated by these sources and retrieve it at a later time, when required. Typically, this is done using components such as batteries, however this project will look at an alternative method which involves using phase change materials (PCMs) to store the energy generated as thermal energy. PCMs take advantage of latent heat to store large amounts of energy without the temperature of the PCM changing. Twisted tapes are commonly used inserts to enhance heat transfer within a tube. Additionally, modified pipe geometry can also be used to increase the heat transfer rate.

Studies such as the one by Mousa et al. [1] have discussed the various phase change materials that optimize energy consumption in the form of energy piles using enclosed

tube containers. Based on the analysis, it was deduced that the inclusion of PCM in piles increased the charging and discharging capacity, as well as the storage efficiency of the piles. Additionally, the PCM increased the thermal response of concrete during the heating and cooling stages. Lastly, increasing the flow rate did not have a significant effect on the percentage of energy stored and released, relatively to the flow rate increasing percentage.

Pereira da Cunha and Eames [2] looked at phase change materials (PCMs) with phase transitions between 0 and 250 °C. The paper found that for low heating rates (<20 kW) it was beneficial to use complex heat exchanger configurations, but for large power requirements the ideal configuration is large parallel tubes. Another method to increase the heat transfer area was to use encapsulated PCMs. Su et al. [3] looked at various types of solid-liquid phase change materials (PCM) and their advantages and disadvantages. The three classes of PCMs that were considered were organic PCMS, inorganic PCMS and eutectic PCMs. The benefits of paraffins are that they are safe, reliable, predicable, cost effective, non-corrosive and have low vapour pressure. Inorganic compounds were determined to be unsuitable for application and additionally harmful to the environment and human health. The advantages of eutectic PCMs are that their melting temperatures can be adjusted by modifying the ratio of the mixture, they have high thermal conductivity, and they are not susceptible to segregation and supercooling.

Ponnada et al. [4] looked at three design modifications for twisted tapes. They were perforated twisted tapes with alternate axis (PATT), perforated twisted tapes (PTT) and regular twisted tapes (TT). It was determined that PATT was the best overall performer, but due to the increased difficulty in manufacturing associated with the geometry that PTT was an acceptable alternative. The results also suggest the samples with the TR of 3 perform better than those with TRs of 4 and 5. Piritarungrod et al. [5] looked at improving the performance of twisted tapes by adding a narrowing taper along the direction of the flow. The two parameters that were varied as part of the study were the twist ratio and the taper angle. It was determined that the top performer of all the designs that were studied was the one that combined the 0.9° taper and the 3.5 twist ratio.

Promvonge et al. [6] looked at improving the heat transfer in a pipe by the addition of rib features to the ID of the pipe and adding two twisted tape inserts into the pipe. It was determined that the ideal twist ratio for the double twisted tape inserts was 8. This had a significant impact as the thermal performance enhancement factor when a twist ratio of 8 is used is approximately 2 to 2.5 times that when a twist ratio of 2 is used. Zheng, Xie and Zhang [7] looked at improving the performance of twisted tape inserts by adding dimple features and protrusion to them. The dimples performed slightly better than the protrusion as they led to an increase of about 25% compared to that of 20% for the protrusion. Additionally, the impact of nanofluids was also analyzed. The addition of nanofluid led to an increase of heat transfer coefficient by up to 60%.

Thirumaniraj [8] looked at designing and analyzing an efficient thermal energy storage (TES) system using paraffin wax as the phase change material (PCM). The paraffin wax was encased in stainless steel balls, that were placed throughout a TES stainless steel tank that was fabricated for this experiment. Through calculating the heat given by hot water as well as the heat gained by cold water using data collected, an efficiency calculation of 62.22% was obtained for the thermal storage system. ANSYS modeling was also performed to show visual and numerical results for the heat exchange found through charging and discharging processes. Sadhishkumar [9] explored the use of phase change material (PCM) in a thermal energy storage (TES) unit. The use of paraffin as the PCM was explored. Water was used as the heat transfer fluid. It was found that that the studied storage system gave a better performance than the conventional solar water heating system.

Lim [10] explored the efficacy of twisted tapes to enhance convection heat transfer using different twist ratios in laminar flow. Results showed that twisted tapes increased the friction factor up to 10 times, and Nusselt number up to 3 times, with these values increasing as twist ratio was decreased. Saysroy [11] investigated the thermal and fluid behaviors of tubes with multi-channel twisted tapes at both laminar and turbulent flow

conditions. For laminar flow, it was found that the thermal performance factor increases with an increasing number of channels for Re $\leqslant$ 1200 and decreasing number of channels for 1200 < Re $\leqslant$ 2000. Arunachalam [12] performed experimental studies on the convective heat transfer and friction factor in laminar flow conditions using a straight circular tube with and without V-cut twisted tape inserts with $Al_2O_3$-Cu/watery hybrid nanofluid as the working fluid. The addition of nanoparticles as well as the swirl flow generated by the V-cut twisted tape increases the heat transfer coefficient.

Richardson and Woods [13] focused on the ability of Phase change materials to be used in everyday buildings to increase the thermal mass of the building. It was found that melting point of the PCM needs to be within a certain range to work properly which may lead to this concept only being usable for half of the year which poses a large issue for the other half of the year where the PCM is effectively useless. Soares et al. [14] provided an in depth look at all the potential uses of PCM when focusing on buildings. It takes the research that has been performed on the topic from other authors and summarizes the information following each step and consideration of utilizing PCM's in a building structure.

Hosseinejad et al. [15] reviewed numerical results when analyzing a twisted tape and its physical properties within turbulent water flow regimes. For this analysis two twisted tapes are placed side by side with one instance where they are aligned and another where they are unaligned. When comparing the two orientations of the tapes, the unaligned regime led to better heat transfer. Song et al. [16] analyzed microencapsulated phase change materials (MPCMs) and how they impact the use within a slurry and the use of different twisted tapes. It was observed that when comparing the friction factors of the MPCM slurry with different twist ratios, it is evident that the lower the twist ratio is, the higher the friction factor becomes.

Waqas and Uddin [17] explored the idea of free cooling as an alternative to compressor-based air conditioning and the idea of phase change material, which allows for latent heat storage. It was determined that PCMs with high thermal conductivity, increased system performance and resulted in the discharging and charging of the PCM within a limited time. In the paper by Tao et al. [18], the lattice Boltzmann method is used to conduct the performance of latent heat storage (LHS) on metal form and paraffin composite phase change materials (CPCM). It was found that the ideal CPCM should have a porosity of 0.94 and PPI of 45. Hariharan et al. [19] looked at the melting and solidification behavior of paraffin phase change material which is encapsulated in a stainless-steel sphere. It was found that the solidification process is faster than the melting process due to higher thermal conductivity of the solid PCM used.

Gorjaei and Shahidian [20] looked at how the twisted tape insert and nanofluid turbulent flow may increase the heat transfer in a curved tube. From the analysis, it was concluded that the implementation of the twisted tape insert led to a turbulence in the boundary layer, which led to increase in convective heat transfer coefficient. In Khoshvaght-Aliabadi and Eskandari's [21] paper, the heat transfer and overall performance of twisted-tape inserts are analyzed for different twist lengths under Cu-water nanofluid. It was found that all twisted tapes with non-uniform length had higher heat transfer coefficient and Nusselt number values than uniform twisted tape length. An article by K. Papazian et al. [22] looks at thermal performance and efficiency of a circular pipe with two inserts, one with a porous medium with porosity of 0.91, and the other a single twist solid insert. It was determined that the twisted tape insert increases thermal efficiency than the porous media insert.

An article by Liaw et al. [23] looked at the heat transfer of turbulent flow in a helical tube consisting of a twisted tape insert with constant wall temperature. It was concluded that the inclusion of the twisted tape inserts with the helical tube had a higher heat transfer performance. Additionally, it was determined that as the inlet Reynolds number increased, the heat transfer also increased proportionally whereas the friction coefficient dropped with decreasing amount of decrement. Ali et al. [24] looked at the pressure drop and heat

transfer characteristics for a smooth tube and internal helically micro-finned tubes with two different fins to fin height ratios, using water as the working fluid. It was determined that the thermal performance enhanced for the helically finned tube for a range of Reynolds numbers, but with increased pressure drops relative to the smooth tube. Additionally, helically finned tubes with alternating fin height showed a significant decrease in friction factor, while having a significantly small decrease in heat transfer coefficient, compared to the equal fin height tube. In the paper by Kumar et al. [25], the pressure drop and heat transfer characteristics of micro-fin helically coiled tubes were investigated. It was found that under the same operating conditions, the pressure drop and Nusselt number increased with increasing fin and Reynolds number. Additionally, the helical tube coil pitches had a small effect on Nusselt number. Furthermore, the performance factor of a micro-fin helical tube with 8 fins declined, as the coil diameter increased.

In the present study, different inserts inside the pipe were investigated. The aim is to determine the insert that leads to the highest heat enhancement or in another term the performance evaluation criterion. Additionally, the effect of various shaped pipes was investigated. Both PCMs and twisted tapes/varying pipe shapes have been used in various activities, but there is a lack of understanding on how combining these things together will work and what benefits may exist from doing so. This paper will further investigate these factors. Section 2.1 provides the problem description. Section 2.2 presents the finite element formulation. Section 3 presents the results of the analysis. Section 4 is the conclusion of the paper.

## 2. Materials and Methods

### 2.1. Problem Description

In this study we presented a pipe encased by phase change materials. Hot water with a constant inlet temperature of 50 °C flows into the pipe and exchanges heat with the phase change material. The flow of the water at the inlet will have a constant Reynolds number that will ensure laminar flow. The pipe is made of aluminum, has an inner diameter of 1 cm, a wall thickness of 0.3 cm and is 32 cm long. The pipe is protruding at both ends by 1 cm. The phase change materials used in our analysis is paraffin wax with a melting temperature of between 22–26 °C. The phase change material will also be encased by an aluminum case with an overall dimension of 4 × 4 × 30 cm and wall thickness of 0.3 cm. Additionally, all inserts are made with aluminum material. Figure 1 shows the setup of the analysis.

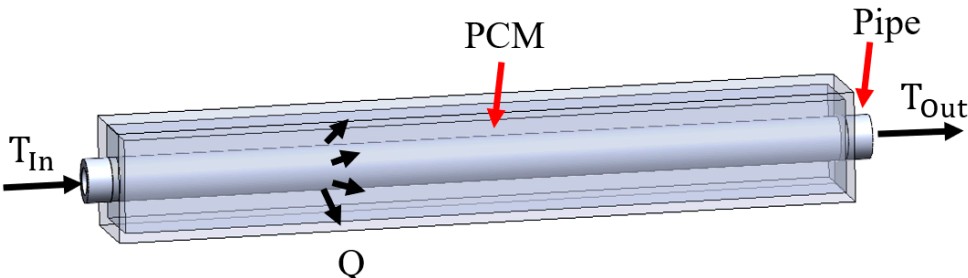

**Figure 1.** Analysis configuration.

The five configurations which are listed in Figures 2 and 3 will be compared to the reference straight pipe with no twisted tape insert. As shown in Figures 2 and 3, five different cases will be investigated. A straight pipe including a twisted tape with a full twist as shown in Figure 2a, then a straight pipe including a twisted tape with a half-twist as presented in Figure 2b. A straight pipe including a twisted tape with a twist ratio of 3, where the twist ratio is the ratio of the length of one-half revolution L divided by the width of the twisted tape W, is shown in Figure 2c. A straight pipe including a twisted tape that is half the length of the rest is shown Figure 2d. Finally, a helical pipe is presented

in Figure 3. Further details regarding the twisted tapes and helical pipe are given in Sections 2.5 and 2.6 respectively.

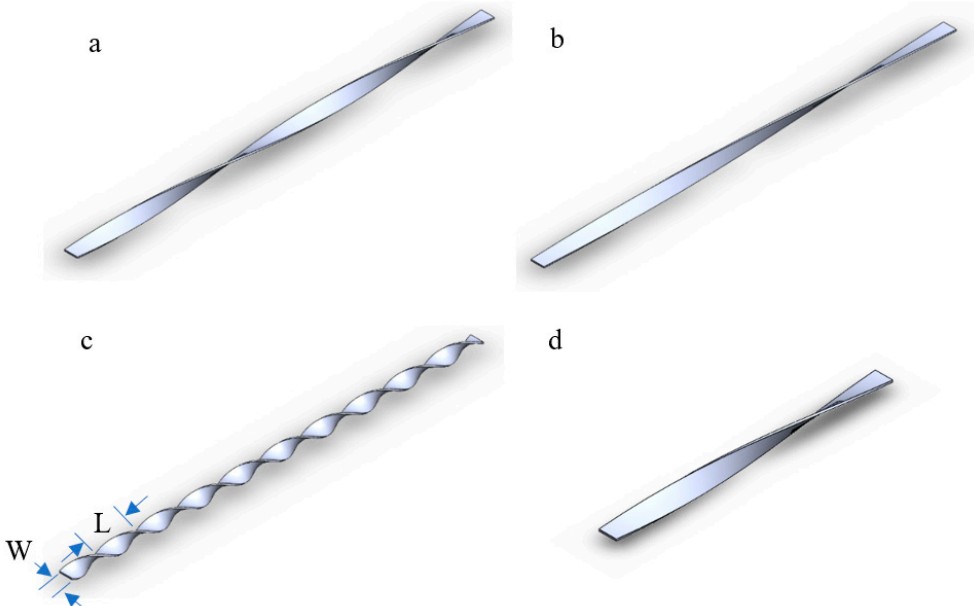

**Figure 2.** (**a**) Full twist twisted tape, (**b**) half-twist twisted tape, (**c**) twisted tape with twist ratio of 3 and (**d**) half-length twisted tape.

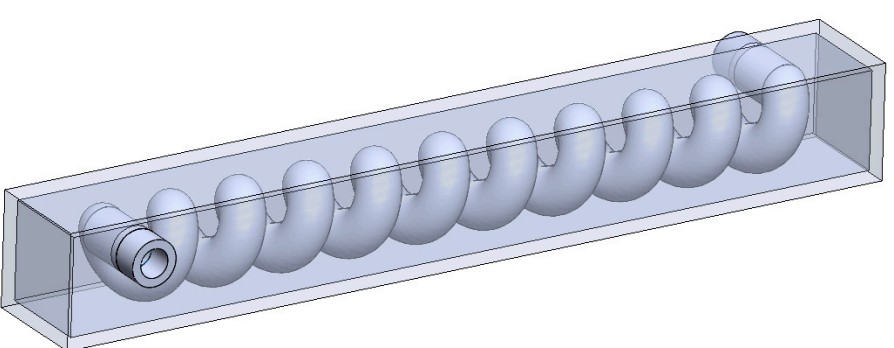

**Figure 3.** Helical pipe configuration.

### 2.2. Finite Element Formulaton

The fluid flow and heat transfer analysis presented in the paper requires solving the full Navier–Stokes equation, continuity equation and the energy equation. COMSOL CFD analysis software was used to complete the analysis using the finite element method. A three-dimensional model has been created and three velocity vectors were added u, v, w, in x, y, z directions, respectively. For the free flow in the model, the formulation adopted are as follows:

x-direction momentum equation:

$$\rho_f \left( u\frac{\partial u}{\partial x} + v\frac{\partial u}{\partial y} + w\frac{\partial u}{\partial z} \right) = -\frac{\partial p}{\partial x} + \mu_f \left( \frac{\partial^2 u}{\partial x^2} + \frac{\partial^2 u}{\partial y^2} + \frac{\partial^2 u}{\partial z^2} \right) \tag{1}$$

y-direction momentum equation:

$$\rho_f \left( u\frac{\partial v}{\partial x} + v\frac{\partial v}{\partial y} + w\frac{\partial v}{\partial z} \right) = -\frac{\partial p}{\partial y} + \mu_f \left( \frac{\partial^2 v}{\partial x^2} + \frac{\partial^2 v}{\partial y^2} + \frac{\partial^2 v}{\partial z^2} \right) + \rho_f g \tag{2}$$

z-direction momentum equation:

$$\rho_f \left( u\frac{\partial w}{\partial x} + v\frac{\partial w}{\partial y} + w\frac{\partial w}{\partial z} \right) = -\frac{\partial p}{\partial z} + \mu_f \left( \frac{\partial^2 w}{\partial x^2} + \frac{\partial^2 w}{\partial y^2} + \frac{\partial^2 w}{\partial z^2} \right) \tag{3}$$

Continuity equation:

$$\left( \frac{\partial u}{\partial x} + \frac{\partial v}{\partial y} + \frac{\partial w}{\partial z} \right) = 0 \tag{4}$$

Energy conservation equation:

$$(\rho Cp)_f \left( u\frac{\partial T}{\partial x} + v\frac{\partial T}{\partial y} + w\frac{\partial T}{\partial z} \right) = k_f \left( \frac{\partial^2 T}{\partial x^2} + \frac{\partial^2 T}{\partial y^2} + \frac{\partial^2 T}{\partial z^2} \right) \tag{5}$$

The dynamic viscosity is represented by $\mu_f$ and the density of the fluid is represented by $\rho_f$. Pressure is represented as p, and gravity vector is notated as g. Velocities are represented by u, v and w and are in the x, y and z dimensions respectively. The specific heat of the fluid is represented by $Cp_f$ and its thermal conductivity is represented by $k_f$. To analyze heat transfer within the aluminum pipe the heat conduction formulation is used.

Additionally, the following equations will be required to model the phase change material. Note in the following equations $\theta_1$ is the fraction of the PCM in the solid phase, $\theta_2$ is the fraction of the PCM in the liquid phase, $\alpha_m$ is the mass fraction, and $L_{1\to2}$ is the latent heat of the phase change material.

$$\rho = \theta_1\rho_1 + \theta_2\rho_2 \tag{6}$$

$$Cp_{PCM} = \frac{1}{\rho}(\theta_1\rho_1 Cp_1 + \theta_2\rho_2 Cp_2) + L_{1\to2}\frac{\partial \alpha_m}{\partial T} \tag{7}$$

$$\alpha_m = \frac{1}{2}\frac{\theta_2\rho_2 - \theta_1\rho_1}{\theta_1\rho_2 + \theta_2\rho_2} \tag{8}$$

$$k = \theta_1 k_1 + \theta_2 k_2 \tag{9}$$

$$\theta_1 + \theta_2 = 1 \tag{10}$$

### 2.3. Boundary Conditions

The temperature and Reynolds number at the inlet will be kept constant at 50 °C and 500 respectively for each configuration that is analyzed. For water in a 1 cm inner diameter pipe this leads to an inlet velocity of 0.0445 m/s. Table 1 shows the physical properties of the materials used in the study.

**Table 1.** Material properties.

| Property | Paraffin Wax | Aluminum | Water |
|---|---|---|---|
| Melting Temperature (°C) | 22–26 | N/A | N/A |
| Thermal Conductivity (W/mK) | 0.2 | 205 | 0.6 |
| Heat of Fusion (kJ/kg) | 100 | N/A | N/A |
| Heat Capacity (J/kgK) | 2100 | 887 | 4200 |
| Density (kg/m$^3$) | 856 (Solid) 778 (Liquid) | 2710 | 1000 |
| Dynamic viscosity (Pa·s) | N/A | N/A | $8.90 \times 10^{-4}$ |
| Ratio of Specific Heats | 1.1 | N/A | 1.0 |

The PCM was modeled in COMSOL using the phase change material module [26]. This allowed for the phase change temperature and transition interval to be set to 22–26 °C.

### 2.4. Mesh Sensitivty

The mesh sensitivity was checked by taking the ratio of the average Nusselt number along the inner diameter of the pipe divided by the accumulated heat. The mesh sizes tested and results for the reference straight pipe configuration are shown in Table 2 and Figure 4. Due to the complexity of the twisted tape features which are also thin bodies, the finer mesh was selected to model the system, to ensure all features were adequately captured for all configurations.

**Table 2.** Mesh information for different levels of meshing.

| Level of Mesh | Information |
| --- | --- |
| Coarse | 22,805 domain elements, 6666 boundary elements, 682 edge elements |
| Normal | 42,631 domain elements, 10,940 boundary elements, 868 edge elements |
| Fine | 72,010 domain elements, 16,104 boundary elements, 1088 edge elements |
| Finer | 165,769 domain elements, 25,616 boundary elements, 1368 edge elements |

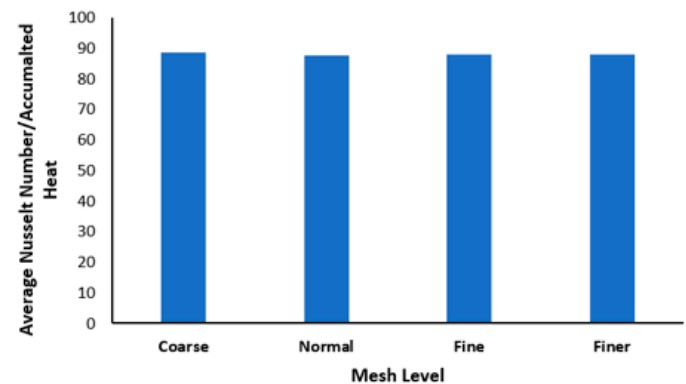

(a) Mesh Sensitivity Analysis

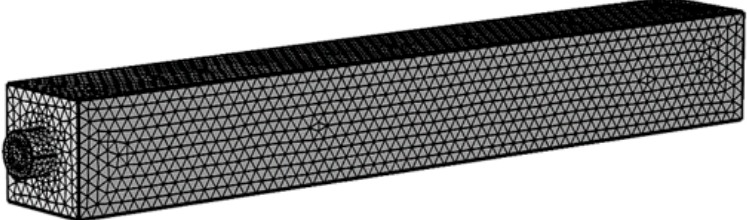

(b) Finite Element Model

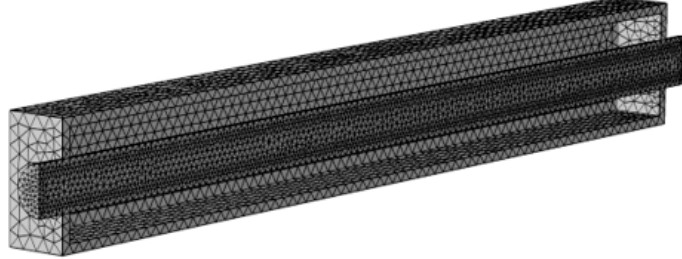

(c) Finite Element Model Cross Section

**Figure 4.** Finite element analysis.

### 2.5. Detailed Dimensions of the Twisted Tape Inserts

To determine the impact of including twisted tapes into the pipe 4 different twisted tapes will be analyzed. Each of the twisted tapes will be 1 cm wide and 0.1 cm thick. All the twisted tapes are 30 cm long besides the half-length twisted tape which is 15 cm

long. Figure 5a shows the dimensions of the full twist twisted tape which completes one revolution over the entire 30 cm length. Figure 5b shows the half-twist twisted tape which completes a half revolution over the entire length. Figure 5c shows the twist ratio 3 twisted tape which completes 5 revolutions over the entire length. Figure 5d shows the half-length twisted tape which has the same twist ratio (length per half revolution divided by width of insert) as the full twist twisted tape but is half the length. The half-length twisted tape will be placed on the inlet side of the pipe.

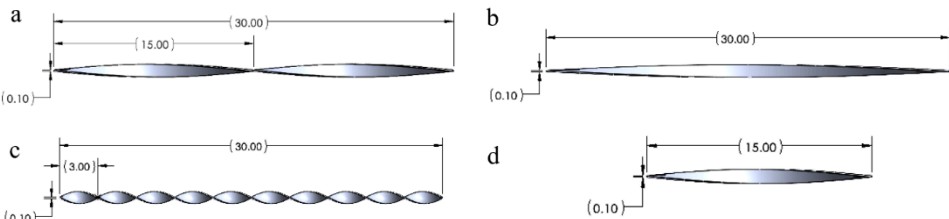

**Figure 5.** Twist ratio of (**a**) full-twist twisted tape; (**b**) half-twist twisted tape; (**c**) twist ratio 3 twisted tape; (**d**) half-length twisted tape.

### 2.6. Detailed Dimensions of the Helical Pipe

The helical pipe will also be made from aluminum, have an inner diameter of 1 cm and wall thickness of 0.3 cm. The helical pipe will have a pitch (i.e., length to complete 1 revolution) of 2.5 cm and a helical diameter of 1.75 cm. The overall length is 25 cm. The helical pipe is shown in Figure 6.

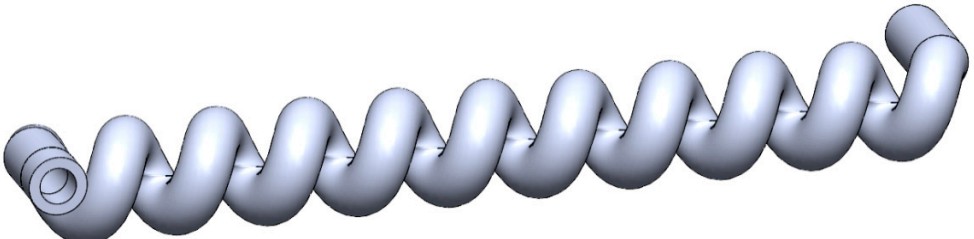

**Figure 6.** Helical pipe configuration.

### 3. Results and Discussion

Determining the best configuration for transferring heat from water flowing in a pipe to phase change materials is the aim of this paper. As previously mentioned, the use of various twisted tapes and a helical pipe will be compared to a reference straight pipe to see what impact they have on the overall performance. Since the straight pipe configuration with no insert will be used as a reference, the performance of the other pipes can be compared to it by taking the ratio of the various factors with respect to the straight pipe.

To determine the inlet velocity of the water, the Reynolds number relationship will be used. To be consistent in comparison, all configurations will be compared using equivalent Reynolds number at the inlet (Re). The Reynolds number will be kept constant at 500 to ensure laminar flow at the inlet of the pipe. The Reynolds number and related inlet velocity can be determined using the following equations.

$$Re = \frac{\rho_f v_{in} D}{\mu_f} \tag{11}$$

$$v_{in} = \frac{Re \mu_f}{\rho_f D} \tag{12}$$

As previously mentioned for water in a 1 cm inner diameter pipe and a Reynolds number of 500 this leads to an inlet velocity of 0.0445 m/s.

To analyze the performance of the pipe several factors will be considered. They are the heat transfer rate of the system ($\dot{Q}$), the Nusselt number (Nu), the friction factor (f), and the performance evaluation criterion (PEC). Further details of these parameters are given in the following sections.

### 3.1. Heat Transfer Rate

The first key parameter that will be looked at it is the heat transfer from the fluid to the phase change material. This can be determined by using the following equation where '$\dot{Q}$' is the heat transfer rate, '$\dot{m}$' is the mass flow rate of the water $Cp_f$ is the heat capacity and '$\Delta T$' is the change of temperature in the water.

$$\dot{Q} = \dot{m}Cp_f\Delta T \tag{13}$$

For a pipe with a constant circular cross section this can be expressed as what is shown in the equation below. In the formula the mass flow rate is replaced by the density of the water ($\rho$), the inlet velocity ($v_{in}$) and the cross-sectional area of the pipe as a function of its diameter (D). It should be noted the average inlet and outlet temperature will be used.

$$\dot{Q} = \frac{\rho_f\pi D^2 v_{in}Cp_f}{4}(T_{in} - T_{out}) \tag{14}$$

The results of Figures 7 and 8 show that all the configurations outperformed the reference straight pipe configuration in terms of the rate of heat transfer rate that they can achieve. The helical pipe configuration was able to nearly double the heat transfer rate of the reference configuration at certain points. It should be noted that after about 600 s the heat transfer rate of the helical pipe configuration drops significantly lower than that of the reference configuration. The reason that this happens is that the helical pipe configuration reaches its storage capacity quicker than the reference configuration. As the storage capacity is reached the heat transfer rate drops significantly since the temperature of the PCM is approximately the same as the water flowing in the pipe. The fact that this happens earlier for the helical pipe is a positive indicator for that configuration as it indicates this configuration can charge quicker than the other configurations.

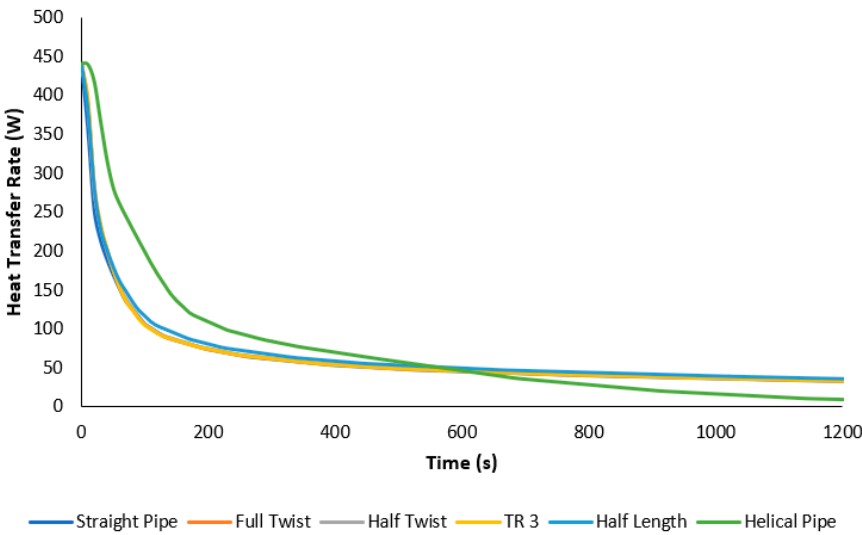

**Figure 7.** Heat transfer rate of the various configurations (note: the full twist, half twist and TR3 lines approximately overlap and some of the lines are not visible).

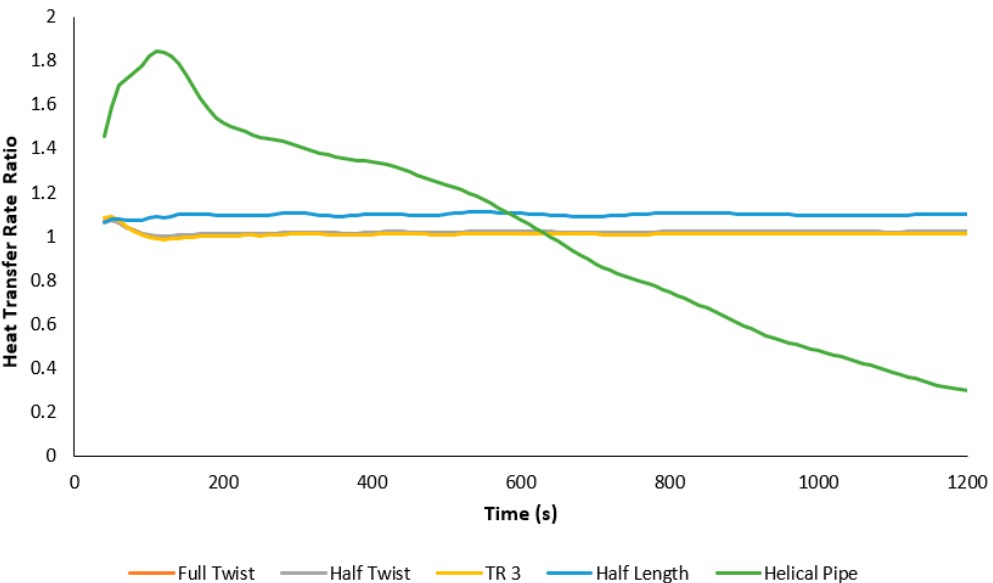

**Figure 8.** Heat transfer rate of the various configurations relative to the straight pipe (note: the full twist, half twist and TR3 lines approximately overlap and some of the lines are not visible).

*3.2. Nusselt Number*

The Nusselt number will be used to determine the ratio between the conductive heat transfer and convective heat transfer of the various configurations. Higher Nusselt numbers indicate an increase in the proportion of heat transfer due to convection. The Nusselt number is given by the following equation.

$$Nu = \frac{hD}{k_f} \tag{15}$$

The heat transfer coefficient will be determined by analyzing the heat transfer across the inner diameter of the pipe normal to the pipe diameter ($Q_{ID}$), the average temperature of the pipe inner wall ($T_{ID}$) and the average temperature of the water. The equation is as follows.

$$h = \frac{Q_{ID}}{T_{ID} - T_f} \tag{16}$$

The Nusselt number for each of the configurations also outperformed the reference configuration as shown in Figures 9 and 10. This is expected as the twisted tape and helical pipe configurations were considered as they are known to encourage the fluid to mix, thus increasing the convection inside the fluid. Out of all the configurations the twisted tape with the highest twist ratio (TR 3 configuration) led to the largest increase in the Nusselt number. The Nusselt number of that configuration was over 50% higher than that of the reference configuration. This again was expected since larger twist ratios for the inserts will force more mixing to occur and thus increase the Nusselt number.

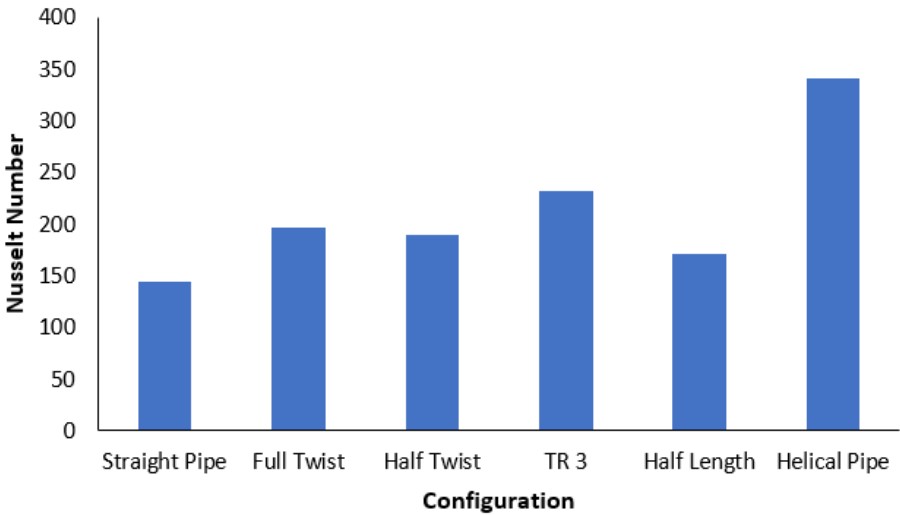

**Figure 9.** Nusselt numbers of the various configurations.

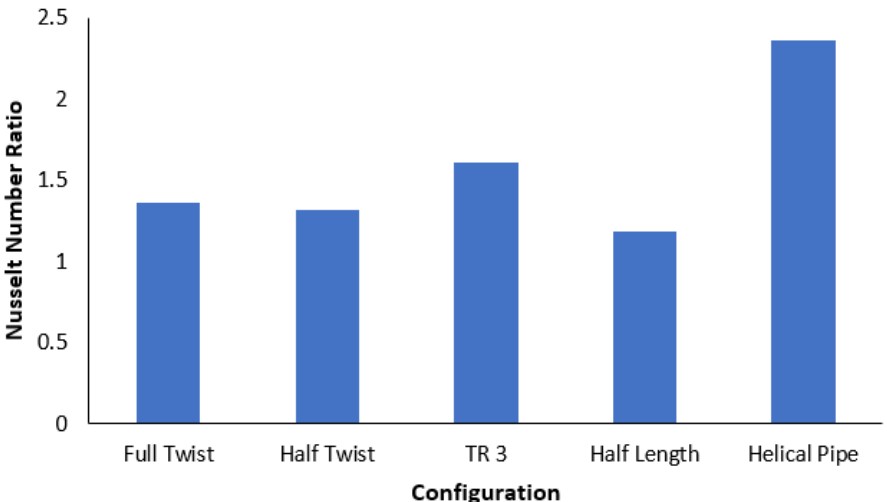

**Figure 10.** Nusselt numbers of the various configurations compared to the straight pipe.

*3.3. Friction Factor*

Since each of the configurations will lead to mixing of the water, the pipe friction factor and the pressure drop need to be investigated. To account for the impact of the increase in pressure drop the friction factor will be analyzed. The friction factor is determined by the following equation.

$$f = \frac{4\Delta P}{\left(\frac{L}{D}\right)\rho_f v_{in}^2} \tag{17}$$

As seen in Figure 11 a draw back for all the configurations is that their friction factors are larger than that of the reference straight pipe. Some of the configurations had a friction factor that is nearly an order of magnitude greater than that of the straight pipe configuration. This will impact the amount of power it takes to pump the fluid through the pipe, with larger friction factors having a more detrimental effect. It was expected that the configurations would have a larger friction factor than the reference configuration as the addition of the helical pipe and twisted tapes impacts the direct flow that is present in the reference straight pipe configuration.

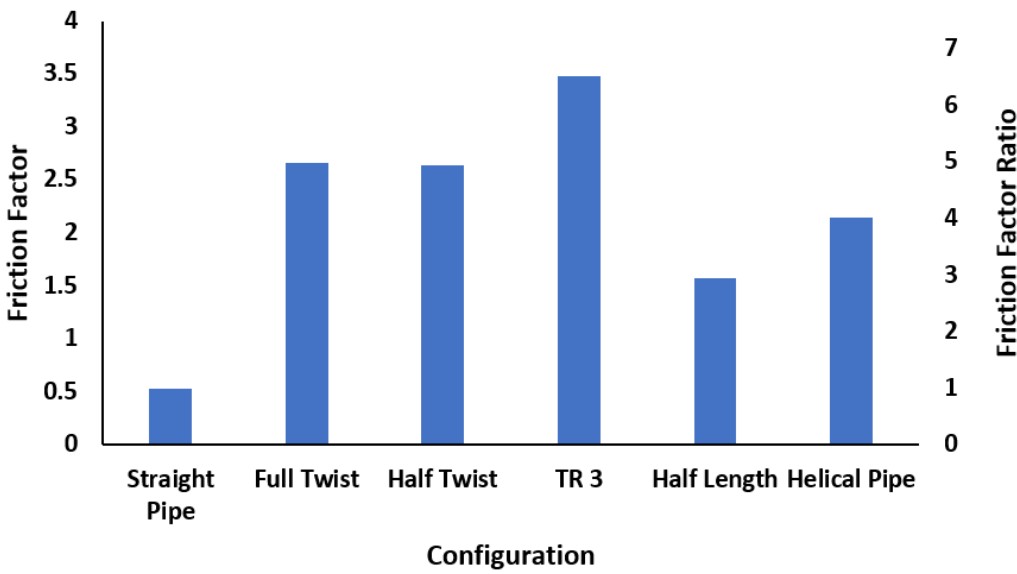

**Figure 11.** Friction factor of the various configurations.

### 3.4. Performance Evaluation Criterion (PEC)

To determine the overall impact of the effects on the Nusselt number and friction factor a performance evaluation criterion (PEC) will be used. The performance coefficient is determined by the following equation.

$$PEC = \frac{NuL}{fD} \tag{18}$$

The left axis of Figure 12 shows the PEC of the various configurations. The helical pipe configuration was the only configuration that was able to outperform the reference configuration. This was due to that fact that for configurations with the twisted tapes, the increase in the friction factor outweighed the gain in the Nusselt number. One may compare the performance of different configuration with a straight pipe by studying the ratio of the PEC for different configurations over the PEC of the straight pipe configuration. The right axis Figure 12 shows the ratio which indicate that the helical pipe is the best design pipe for heat enhancement.

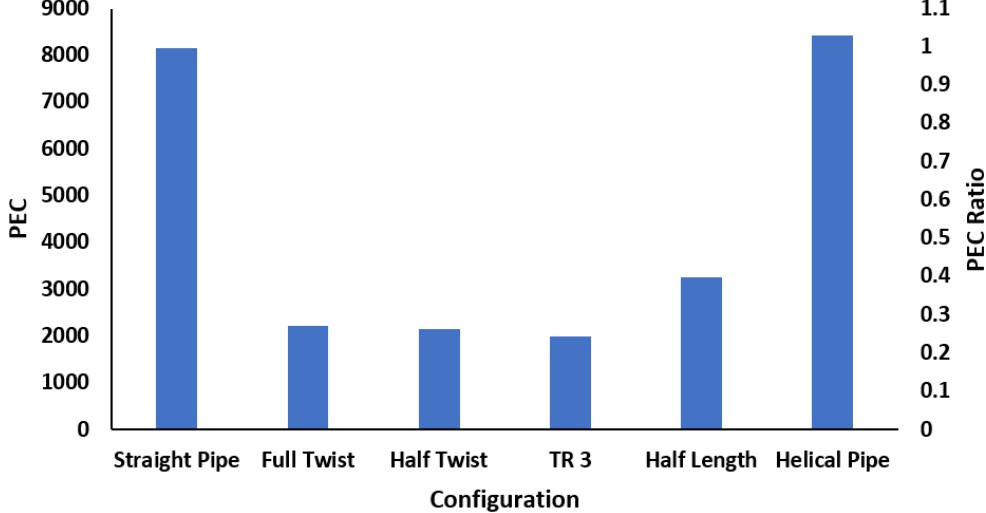

**Figure 12.** PEC of the various configurations.

### 3.5. Effect of Pipe Diameter

To further enhance the performance, the effect of the pipe diameter is studied in detail. In particular, the performance evaluation criterion of the configurations can be further enhanced. To do this, the equation for the Nusselt number and the equation for the friction factor can be directly substituted into the formula for the PEC as is shown below. The PEC of the changing diameter will be notated as PEC$_D$.

$$PEC_D = \frac{\rho_f v_{in}^2 hL^2}{4\Delta P k_f D} \tag{19}$$

If a constant Reynolds number at the inlet is maintained (Re of 500) then the equation relating velocity to Reynolds number (Equation (7)) can be substituted into the equation.

$$PEC_D = \frac{Re^2 \mu_f^2 hL^2}{4\rho_f \Delta P k_f D^3} \tag{20}$$

Splitting the above equation into constant and non-constant parts gives the following.

$$PEC_D = \left(\frac{Re^2 \mu_f^2}{4\rho_f k_f}\right)\left(\frac{hL^2}{\Delta P D^3}\right) \tag{21}$$

The only parameters that can be changed directly in the configuration are the path length of the pipe (L) and the inner diameter of the pipe D (diameter of the pipe) as all components are either constant or implicitly determined by these two parameters. Based on the equation the most effective way to increase the PEC seems to be lowering the inner diameter of the pipe as the performance evaluation criterion is inversely proportional to the cube of the inner diameter of the pipe (without considering what indirect effects the pipe inner diameter has on the pressure drop and heat transfer coefficient). The inner diameter of the pipe that was simulated previously was 1 cm. Two additional simulations were investigated changing the pipe ID to 0.75 and 0.5 cm respectively for a total of three data points. To keep the Reynolds number at the inlet constant, the inlet velocity will have to be determined for these new configurations using Equation (7). Besides those changes all other parameters of the configuration will remain the same as previously specified. The varying parameters of the 3 helical pipe configurations are shown in Table 3.

**Table 3.** Inlet Velocity.

| Pipe ID (cm) | Inlet Velocity (m/s) |
| --- | --- |
| 1 | 0.0445 |
| 0.75 | 0.0593 |
| 0.5 | 0.0890 |

The results of the simulations are shown in Figures 13–15. Overall, as the inner diameter of the pipe is made smaller, the PEC of the system rises. Additionally, the outlet temperature of the outlet water remains cooler for longer which indicates a higher percentage of the heat is being collected from the water as it flows past the PCM. This is likely due to the fact that lowering the pipe inner diameter, while holding the Reynolds number constant, lowers the total mass flow rate since the cross-sectional area of the pipe decreases faster than the inlet velocity increases. This is the case since area is related to the square of the diameter whereas the inlet velocity for a constant Reynold's number is inversely related to the diameter. In other words, as the diameter becomes smaller less heat enters the system, due to a lower mass flow rate, but a higher percentage of the heat that does enter the system is stored in the PCM. This is a limiting factor on how small the diameter can become. The reduction in mass flow rate also has the effect of lengthening the time required for the system to reach its maximum capacity as can be seen by the fact that

the configuration with the 1 cm pipe diameter reaches a heat transfer rate of about zero the earliest in Figure 15.

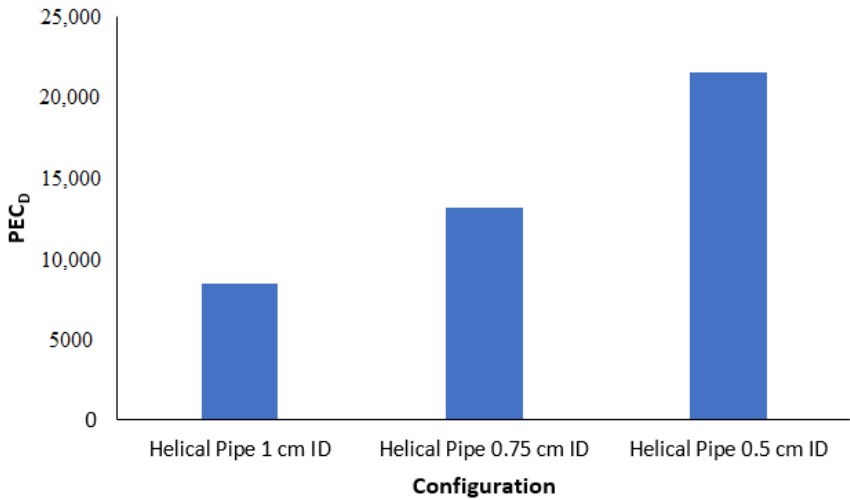

**Figure 13.** PEC$_D$ of the helical pipe configuration with various IDs.

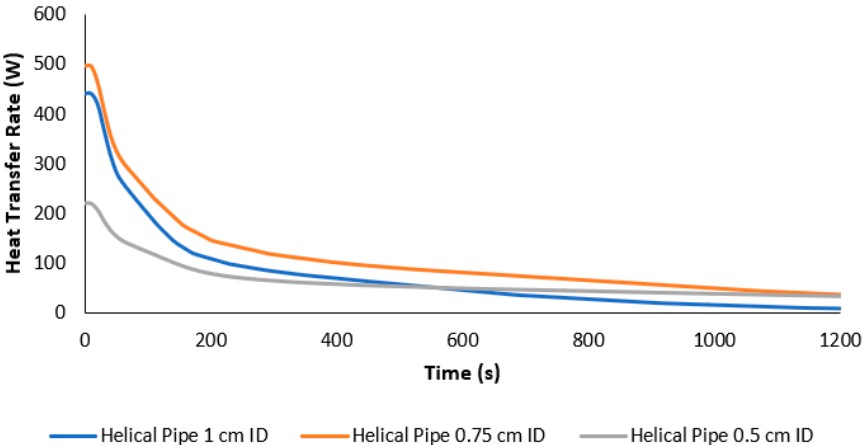

**Figure 14.** Outlet temperature vs. time for the helical pipe configuration with various IDs.

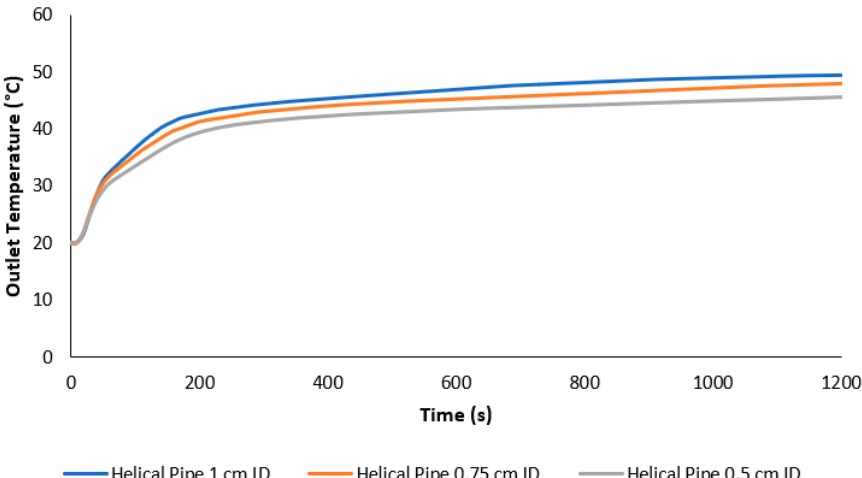

**Figure 15.** Heat transfer rate vs. time for the helical pipe configuration with various IDs.

## 4. Conclusions

We have investigated an approach for storage of heat energy using phase change materials and increasing the heat transfer rate in fluids by incorporating various pipe shapes and twisted tape inserts. It was found that these factors had been previously implemented for separate purposes, but there was a lack of research on the combination of these things.

Several configurations to determine which had the best overall performance were studied. The considered configurations involved either using a helical pipe or adding a twisted tape insert to enhance the overall heat transfer to the phase change material. Paraffin wax was selected as the phase change material. All the initial configurations had common parameters such as overall size, pipe diameter, pipe thickness and thickness of the casing so the performance under similar conditions could be compared. COMSOL was used to simulate the performance of the configurations. The simulation looked at both the fluid flow within the pipe and heat transfer between the water and PCM.

It was determined that both the twisted tapes and helical pipe were able to increase both the overall heat transfer rate and Nusselt number when compared to the straight pipe reference configuration. The twisted tape with the highest twist ratio produced the highest Nusselt number whereas the helical pipe configuration created the highest overall heat transfer rate. When looking at the PEC the helical pipe configuration was the only one that outperformed the reference straight pipe. This was due to the fact that all of the twisted tape inserts significantly increased the friction factor and that outweighed the gain in the Nusselt number.

Due to the above factors, it was determined to only continue analyzing the helical pipe configuration. To further enhance the helical pipe configuration, analysis of the parameters influencing the PEC was done. It was determined that lowering the pipe inner diameter would likely increase the PEC of the configuration. To confirm this hypothesis two additional simulations were done where only the pipe inner diameter and inlet velocity (to maintain a constant Reynold's number) were changed. These inner diameters that were analyzed were 1, 0.75, and 0.5 cm, and the simulations confirmed the hypothesis. Lowering the pipe inner diameter to 0.75 cm increased the PEC by approximately 50% compared to that of a 1 cm inner diameter, whereas lowering the inner diameter to 0.5 cm increased the PEC by approximately 150%. Additionally, the resulting lower mass flow rates caused by the smaller inner diameters led to the thermal batteries taking longer to reach their maximum capacity.

Overall, the helical pipe configuration with a 0.5 cm pipe inner diameter had the highest PEC. It is recommended that further research be done on how to further enhance the design. Possible design modifications include changing the type of phase change material that is used, the Reynolds number of the flow at the inlet, the path length (i.e., pitch and helical diameter) of the helical pipe and varying the inlet temperature.

**Author Contributions:** Conceptualization, M.Z.S.; methodology, D.A., S.A.-B., A.M. and J.P.; software, D.A.; validation, S.A.-B. and A.M. and J.P.; formal analysis, D.A.; investigation, D.A., S.A.-B., A.M. and J.P.; resources, M.Z.S.; data curation, D.A.; writing—original draft preparation, D.A., S.A.-B., A.M. and J.P.; writing—review and editing, M.Z.S.; visualization, D.A. and S.A.-B.; supervision, M.Z.S.; project administration, M.Z.S.; funding acquisition, M.Z.S. All authors have read and agreed to the published version of the manuscript.

**Funding:** This research was funded by [National Science and Engineering Research Council Canada, Faculty of Engineering and Architecture, Ryerson University] and [Qatar Foundation] grant number [NPRP12S-0123-190011].

**Data Availability Statement:** Not applicable.

**Conflicts of Interest:** No conflict of interest.

### Nomenclature Symbols

| | |
|---|---|
| Cp | Heat capacity (J/kg·K) |
| D | Diameter (m) |
| f | Friction Factor |
| g | Gravity |
| h | Heat transfer coefficient |
| k | Thermal conductivity |
| l | Length (m) |
| m | Mass (kg) |
| Nu | Nusselt Number |
| P | Pressure |
| PEC | Performance Evaluation Criterion (unitless) |
| Re | Reynolds Number (unitless) |
| T | Temperature (°C) |
| TR | Twist Ratio (unitless) |
| t | Thickness (m) |
| u | Velocity in x direction (m/s) |
| v | Velocity in y direction (m/s) |
| w | Velocity in z direction (m/s) |

### Greek Symbols

| | |
|---|---|
| $\alpha_m$ | Mass fraction (unitless) |
| $\theta_1$ | Proportion of PCM in solid phase (unitless) |
| $\theta_2$ | Proportion of PCM in liquid phase (unitless) |
| $\mu$ | Dynamic viscosity (Pa·s) |
| $\rho$ | Density (kg/m$^3$) |

### Subscripts

| | |
|---|---|
| f | Fluid property |
| ID | At pipe ID |
| in | At pipe inlet |
| out | At pipe outlet |

### Accents

| | |
|---|---|
| $\dot{X}$ | Variable "X" with respect to time |

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
