# Peer review of "Storing Energy from External Power Supplies Using Phase Change Materials and Various Pipe Configurations"

_processes, doi:10.3390/pr9071160_

Round 1

Reviewer 1 Report

The article deals with storing thermal energy using PCMs. The problem presented here is important from the practical point of view. In my opinion, this paper was written in the accordance with the requirements for the scientific reports.

I present my comments below.

In my opinion, the characteristics of the PCM material used for simulation should be presented in this paper (specific heat of the paraffin wax as a function of temperature).

The authors wrote in the article that the melting temperature is between 22-26°C. Please describe how this feature/characteristic was modelled in COMSOL CFD software.

It would be useful to present the mesh in the cross-section of the model in Fig. 4.

There are a lot of variable units missing from the nomenclature.

In conclusion, I must add that the shortcoming of this study is the lack of model validation. The results of the experiment in this type of research may differ significantly from the results of computer simulation.

Author Response

Thank you for taking time to review our paper

Reviewer 2 Report

The paper is written in detail, well explained and is appealing to the readers but needs slight improvement as follows: 

1. The title should be modified to reflect that the work done is on pipe configurations

2. The literature review may be improved by adding recent publications on similar themes, e.g. 
I.   Turbulent convective heat transfer in helical tube with twisted tape insert (2021) 
II.  Numerical Analysis of Heat Transfer and Pressure Drop in Helically Micro-Finned Tubes (2021) 
III. Numerical investigation of heat transfer and pressure drop characteristics in the micro-fin helically coiled tubes (2021)

3. In Mesh sensitivity, if the ratio of Average Nusselt number and accumulated heat transfer is comparable for all the settings then please elaborate why the finer mesh is used?  

4. In Figure 7 only 3 lines are visible but 5 lines are represented and explained, please use clear symbols. Similarly Figure 8 also lacks the presentation of the results obtained. 

5. It has been observed that multiple graphs are presented which can be combined together on a single graph e.g. Friction factor and friction factor ratio can be shown on one graph.

6. Figure 13 shows a very small increase between the reference and helical pipe used so what is the reason to go for the complexity for such a small increase?

7. Page 19 Line 389 typo mistake

8. Please re-analyze Figure 16,  in context of statements on page 15 lines 413-416 and in result section page 17 lines 457-461. If the PEC is increased by reducing the inner diameter and temperature remains cooler for longer, then how does the 0.5cm diameter pipe have less heat transfer rate from the other two mentioned in the graph if the PEC is higher of other configuration?

Author Response

Thank you for taking the time to review our paper
